# Advances in 5-Aminolevulinic Acid Priming to Enhance Plant Tolerance to Abiotic Stress

**DOI:** 10.3390/ijms23020702

**Published:** 2022-01-09

**Authors:** Shuya Tan, Jie Cao, Xinli Xia, Zhonghai Li

**Affiliations:** National Engineering Laboratory for Tree Breeding, College of Biological Sciences and Technology, Beijing Forestry University, Beijing 100083, China; tsy20@bjfu.edu.cn (S.T.); CaoJie@bjfu.edu.cn (J.C.); xiaxl@bjfu.edu.cn (X.X.)

**Keywords:** 5-ALA, defense priming, abiotic stress, multi-omics, plant hormone

## Abstract

Priming is an adaptive strategy that improves plant defenses against biotic and abiotic stresses. Stimuli from chemicals, abiotic cues, and pathogens can trigger the establishment of priming state. Priming with 5-aminolevulinic acid (ALA), a potential plant growth regulator, can enhance plant tolerance to the subsequent abiotic stresses, including salinity, drought, heat, cold, and UV-B. However, the molecular mechanisms underlying the remarkable effects of ALA priming on plant physiology remain to be elucidated. Here, we summarize recent progress made in the stress tolerance conferred by ALA priming in plants and provide the underlying molecular and physiology mechanisms of this phenomenon. Priming with ALA results in changes at the physiological, transcriptional, metabolic, and epigenetic levels, and enhances photosynthesis and antioxidant capacity, as well as nitrogen assimilation, which in turn increases the resistance of abiotic stresses. However, the signaling pathway of ALA, including receptors as well as key components, is currently unknown, which hinders the deeper understanding of the defense priming caused by ALA. In the future, there is an urgent need to reveal the molecular mechanisms by which ALA regulates plant development and enhances plant defense with the help of forward genetics, multi-omics technologies, as well as genome editing technology.

## 1. Introduction

Defense priming refers to a physiological state (a state of readiness for defense) that is induced after the plants perceive a variety of stimuli, such as pathogens, arthropods, and abiotic cues, as well as chemicals (Figure 1). In this state, the defense responses are deployed in a faster, stronger, and/or more sustained manner, thus defense priming is considered an adaptive and low-cost defensive strategy [1]. The pathogen produces molecules with different biochemical natures (peptides, polysaccharides, or lipids) that are sensed by plants through the corresponding receptors, thus inducing plant priming [2]. The common bean (*Phaseolus vulgaris*) activates enhanced plant defense by inoculation with *Rhizobium etli* and develops stronger resistance to *Pseudomonas syringae* compared with unstimulated plants [3]. Herbivore-inducible plant volatiles (HIPVs) released in response to herbivore attack can induce priming in neighboring plants, which exhibit faster or stronger defense activation and insect resistance when subjected to insect feeding [4]. Repeated exposure of plants to mild abiotic signals, such as heat, cold, or salt, can also trigger plants into a defense priming state [5]. In addition, pretreatment with low concentrations of chemicals such as hydrogen peroxide (H_2_O_2_), sodium hydrosulfide (NaHS), sodium chloride (NaCl), sodium nitroprusside (SNP), γ-aminobutyric acid (GABA), melatonin, polyamines (PAs), as well as 5-aminolevulinic acid (ALA) also induces plants into defense priming status [6,7,8], in which plants respond to biological and abiotic stresses through faster and stronger defensive activation [9]. Furthermore, the ability of priming to enhance stress tolerance can be self-propagating. For example, defense priming occurs in roots, while transcriptional differences can be detected in both roots and leaves [10]; mobile wound signals transmitted from local damaged sites to distal undamaged sites induce the whole plant into priming [11]; infestation of plants with phloem-feeding whitefly (*Bemisia tabaci*) triggers local and systemic defense priming [12]. Interestingly, the priming status of plants can be inherited across generations. For example, a transgenerational priming response against pathogen attack can last for at least two generations in common beans upon treatment with the priming agent GABA [13]. This transgenerational inheritance of defense priming may involve epigenetic regulation [2,14].

## 2. Biosynthesis of 5-Aminolevulinic Acid

5-Aminolevulinic acid (ALA) has been found to increase tolerance to various stresses and is a promising chemical molecule for application in agriculture. ALA is widely found in various living organisms, including bacteria, algae, plants, and animals, and is a universal precursor for the synthesis of all tetrapyrroles (chlorophyll, heme, siroheme, vitamin B_12_, and phytochromobilin) [15,16]. Therefore, to better understand the biological role of ALA and underlying molecular mechanisms in defense priming, we firstly made a brief introduction to its biosynthesis (Figure 2). There are two pathways for the biosynthesis of ALA, the C_4_ pathway (or Shemin pathway) and the C_5_ pathway (or Beale pathway) [17]. The C_4_ pathway is found in animals, fungi, and some algae and bacteria. In this pathway, ALA is produced by direct condensation of succinyl-CoA and Gly catalyzed by ALA synthase. The C_5_ pathway is mainly found in plants and archaea, which consists of a three-step enzymatic reaction [18]. Firstly, _L_-glutamate is ligated to tRNA^Glu^, which is catalyzed by glutamyl–tRNA synthetase (GluTS) to form _L_-glutamy–tRNA. Secondly, the carboxyl group of Glu-tRNA is reduced to a formyl group and _L_-Glu-tRNA is converted to _L_-glutamic acid 1-semialdehyde (GSA). GluTR plays a key role during the synthesis pathway of ALA. Lastly, GSA undergoes an isomerization reaction catalyzed by glutamate-1-semialdehyde aminotransferase (GSAT) to form ALA. These reactions are located in the chloroplast stroma [19].

Chlorophyll and heme, with ALA as their precursors, are involved in many biochemical processes. They share a series of steps in the synthesis pathway from ALA to protoporphyrin IX (PpIX). It starts with two molecules of ALA catalyzed by ALA dehydratase (ALAD), which aggregates to form a pyrrole ring called porphobilinogen (PBG); this is followed by a six-step enzymatic reaction in which four molecules of PBG polymerize to form a porphyrin structure, eventually forming PpIX. The synthetic pathway branches off here to produce heme or chlorophyll, respectively. PpIX chelates Fe^2+^ by Ferrochelatase (FECH) to produce heme or chelates Mg^2+^ by Mg-chelatase (MCH) and undergoes a series of catalytic reactions to produce chlorophyll. ALA was originally obtained by chemical methods, which is a complex process and difficult to purify, resulting in a low yield and high price [20]. At present, ALA can be produced commercially through easier, cheaper, and sustainable microbial methods [21,22,23].

## 3. ALA Priming Enhances Plant Resistance to Abiotic Stresses

Abiotic stresses are considered to be major environmental factors limiting the yield and quality of crop plants. Developing effective strategies to mitigate the deleterious effects of abiotic stresses is critical for sustainable agriculture and food security. Recent studies have shown that exogenous treatment of plants with 5-ALA can enhance abiotic stress tolerance by inducing molecular and physiological defense mechanisms, providing a promising strategy for mitigating abiotic stress in plants. Here, we focus on reviewing the physiological and molecular mechanisms of action of ALA priming in alleviating salt stress, extreme temperature stress, drought stress, and UV stress, and construct a regulatory network to show them systematically (Figure 3; Table 1).

### 3.1. ALA Priming Alleviates Salt Stress

Salt stress is one of the most deleterious environmental factors that hamper agricultural productivity worldwide by stimulating osmotic stress, ionic toxicity, nutritional disorders, and oxidative stress simultaneously [24,25]. Salt stress results in an imbalance in intracellular ionic homeostasis, triggering the increase in Na^+^ concentration and the decrease in K^+^ concentration in plant tissues [26]. Priming with ALA increases the transcripts and protein accumulations of SOS1 (Na^+^/H^+^ antiporter) and HA3 (proton pump) on the plasma membrane (PM) as well as NHX1 (Na^+^/H^+^ antiporter) and VHA-A (proton pump) on the vesicle membrane compared with the unprimed cucumber (*Cucumis sativus*) in response to salt stress. The ion transporter proteins SOS1 and NHX1, with the energy provided by proton pump HA3 and VAH-A, help cucumber excrete Na^+^ from the cytoplasm or transfer it to the vesicles, resulting in a high-low-high osmotic potential in the vesicle-protoplast-exosome, and thus alleviating ion toxicity induced by salt stress. Pretreatment with ALA upregulates the expression of high-affinity K^+^ transporter protein 1 (HKT1) that regulates Na^+^/K^+^ homeostasis in cucumber cells and maintains normal metabolic activities in cells under salt stress conditions [27,28]. Proline accumulates in response to salinity and is a common compatible osmolyte in higher plants. Exogenous application of ALA upregulates delta-1-pyrroline-5-carboxylate synthase (P5CS) that controls the rate-limiting step of glutamate-derived proline biosynthesis in Oilseed rape (*Brassica napus*) and enhances tolerance to salt stress [26,29]. In addition, priming with ALA relieves cell oxidation stress caused by salt stress by improving the activity of antioxidant enzymes, including superoxide dismutase (SOD), catalase (CAT), and peroxidase (POD), and promoting the activity of enzymes involved in the ascorbate-glutathione cycle (AsA-GSH), including ascorbic acid oxidase (AAO), ascorbate peroxidase (APX), glutathione reductase (GR), dehydroascorbic acid reductase (DHAR), and monodehydroascorbic acid reductase (MDHAR) [30,31,32,33].

In addition to coping with osmotic stress and oxidative stress caused by salt stress, priming with ALA improves plant salt tolerance by increasing photosynthetic assimilation and promoting nitrogen metabolism. Cassia seed (*Cassia obtusifolia*), peach (*Prunnus persica*), and oilseed rape treated with ALA showed an increase in the net photosynthetic rate (Pn) and transpiration rate (Tr), as well as the photochemical efficiency of photosystem II (Fv/Fm) and the non-photochemical quenching (NPQ) during salt stress [29,31,34]. Application of ALA on the leaves leads to swollen chloroplasts and dilation of thylakoid membrane under salt stress conditions, thereby modifying photosynthetic sites and increasing photosynthetic efficiency [35]. Pretreatment with ALA decreases the expression of ferrochelatase, catalyzing the insertion of Fe^2+^ into protoporphyrin, in Oilseed rape in response to salt stress, thereby inhibiting heme synthesis and increasing chlorophyll synthesis [36]. Additionally, ALA treatment significantly increases the activities of nitrate reductase (NR), glutamine synthetase (GS), glutamate synthase (GOGAT), and glutamate dehydrogenase (GDH) and decreases the activity of nitrite reductase (NiR) in watermelon (*Citrullus lanatus*). This indicates that ALA regulates nitrogen metabolism to alleviate the cellular toxicity caused by the massive accumulation of nitrate and ammonium salts due to salt stress [37].

The induction of salt tolerance in plants by ALA may be achieved through nitric oxide (NO). ALA treatment increased NO and NOS activity in leaves, suggesting that ALA triggers NO synthesis by activating NOS, and thus improves salt tolerance in maize (*Zea mays*) [38]. In supporting this hypothesis, ALA-induced salt tolerance is completely abolished by treatment with a scavenger of NO, 2-4-carboxyphenyl-4,4,5,5-tetramethylimidazoline-1-oxyl-3-oxide (cPTIO), a stable free radical compound that reacts with NO to form an imino nitroxide free radical.

### 3.2. ALA Priming Increases Plant Tolerance to Extreme Temperature

The rate of plant growth and development depends on the temperature surrounding the plant. Considering that plants are sessile, their survival depends on the efficient activation of resistance responses to thermal stress. No organism can withstand the full range of biosphere temperatures, and each species has a specific temperature range represented by minimum, maximum, and optimum temperatures. Extreme temperature events are expected to become more intense and frequent and last longer than those observed in recent years [39]. Extreme temperature adversely affects almost all aspects of plant growth, development, reproduction, and yield [40]. Warm temperatures cause growth restriction, increase the transpiration rate, and damage photosynthetic organs in plants [41], while extreme low temperatures lead to membrane damage, inhibition of photosynthetic properties, and oxidative stress [42]. ALA-pretreated cucumber leaves had higher antioxidant enzyme activity, higher levels of proline and soluble sugar content, and weaker growth inhibition under high-temperature stress conditions [43]. Priming with ALA increases germination and seedling emergence in red pepper (*Capsicum annuum*) and reduces tissue electrolyte leakage in rice (*Oryza sativa*) under cold stress [44,45]. Pretreatment with ALA also increases chlorophyll content and photosynthetic capacity of cucumber and enhances ribulose-1,5-bisphosphate (RuBP) carboxylase activity in maize under cold stress conditions [46,47,48]. Furthermore, ALA treatment also improved the antioxidant capacity of plants in response to cold stress by increasing the activities of SOD, APX, GR, CAT, and heme oxygenase-1 (HO-1) in red pepper, drooping wild ryegrass (*Elymus nutans*), and soybean plants (*Glycine max*) [49,50,51]. Interestingly, ALA priming upregulates the expression levels of *respiratory burst oxidase homologue1* (RBOH1) in tomato (*Solanum lycopersicum*) and leads to the production of H_2_O_2_, which serves as a signaling molecule to activate defense against cold stress [52].

In addition, studies in drooping wild ryegrass and tomato suggest that ALA may directly trigger NO production or indirectly promote NO production by inducing jasmonic acid (JA) and H_2_O_2_ [53,54]. NO activates antioxidant enzyme activity and PM H^+^-ATPase and maintains Na^+^/K^+^ homeostasis, thereby reducing cold stress-induced injury [55]. Moreover, priming with ALA induces the expressions of genes involved in the biosynthesis of PA in rice [7], which enhances tolerance to cold stress [56]. Besides, studies in cucumber have shown that ALA priming increases tolerance to cold stress by regulating the biosynthesis of classic phytohormones such as JA, indole-3-acetic acid (IAA), brassinosteroid (BR), cytokinins (CKs), gibberellin (GA4), and abscisic acid (ABA) during cold stress [47]. In the future, further studies are needed to investigate how ALA regulates phytohormone synthesis under cold stress conditions and whether it regulates key components of the phytohormone signaling pathway so as to further reveal the molecular mechanisms by which ALA enhances low-temperature tolerance in plants.

### 3.3. ALA Priming Mitigates Drought-Induced Damage

Under natural and agricultural conditions, plants are subject to various environmental stresses during growth and development. Among them, drought is one of the most severe environmental stresses, which occurs as a result of temperature dynamics, light intensity, and low rainfall and affects plant biomass production, quality, and energy. Drought stress limits photosynthesis in plants by causing stomatal closure and reduced water content, as well as leading to excessive production of reactive oxygen species, which can inhibit plant growth [57]. ALA pretreatment can maintain moisture in the seedlings of oilseed rape and Kentucky bluegrass (*Poa pratensis*), thus enhancing leaf relative water content (RWC) [58,59]. It can also increase the contents of proline and foliar N in wheat (*Triticum aestivum*), as well as Ca^2+^ in the roots under drought conditions [60,61,62]. In addition, in studies with Kentucky bluegrass and sunflower (*Helianthus annuus*), priming with ALA increases the activities of antioxidant enzymes such as catalase (CAT), superoxide dismutase (SOD), ascorbate peroxidase (APX), and glutathione reductase (GR), which reduce the production of ROS, including H_2_O_2_ content and O_2_^•−^ production, thereby improving tolerance against drought stress [58,63]. Priming with ALA also preserves plant photosynthesis in oilseed rape, wheat, and sunflower by suppressing chlorophyll degradation and increasing photosynthetic rate (Pn) during drought stress [59,61,64,65]. Furthermore, pretreatment with ALA induces the expressions of enzymes involved in the Calvin cycle such as triose-3-phosphate isomerase (TPI) and fructose-1,6-bisphosphate aldolase (FBPA) [66]. Interestingly, in addition to enhance the drought resistance of plants, ALA priming also improves waterlogging tolerance in Fig (*Ficus carica*), with higher levels of antioxidant enzyme activity, photosynthetic efficiency, and root respiration [67].

### 3.4. ALA Priming Attenuates UV-B-Induced Damage

Ultraviolet-B (UV-B) radiation is a component of sunlight that induces several plant photomorphogenic responses, including hypocotyl growth inhibition and cotyledon curling [68]. High-intensity UV-B injures plants by damaging DNA, impaired photosynthesis, and cell death, and triggering the accumulation of ROS [69]. Priming with ALA was reported to significantly reduce plant damage from UV-B radiation by promoting photosynthesis, enhancing antioxidant capacity, and improving nitrogen metabolism. As a key precursor of chlorophyll biosynthesis, ALA alleviated the deficiency of chlorophyll biosynthesis during UV-B stress; ALA pretreatment upregulates the expression of genes involved in chlorophyll biosynthesis such as glutamyl-tRNA reductase (HEMA1), Mg-chelatase (CHLH), and protochlorophyllide oxidoreductase (POR) in pigeon pea (*Cajanus cajan*), thus promoting plant photosynthesis during UV-B stress [8,70]. In addition, ALA priming-increased activities of antioxidant enzymes are essential for lettuce (*Lactuca sativa*) resistance to UV-B stress [71]. In addition to enzymatic antioxidants, ALA also increases the content of non-enzymatic antioxidants such as flavonoids and phenolics [8]. Under UV-B stress conditions, ALA priming significantly improves the activities of nitrate reductase (NR), nitrite reductase (NiR), glutamine synthetase (GS), and glutamate synthase (GOGAT), and then increases the levels of NO_3_^−^ and NO_2_^−^ in the seedlings of pigeon pea [70]. Collectively, ALA priming contributes to UV-B tolerance by regulating photosynthesis, antioxidant, and nitrogen metabolism in plants.

## 4. Application of ALA in Agriculture and Medicine

ALA is a growth regulator that regulates the growth and development of plants at different growth stages. Soaking seeds with ALA reduces the germination time and increases the germination index of pigeon pea, and promotes *Arabidopsis thaliana* root elongation by regulating auxin transport [8,72]. ALA significantly increased yields through root or foliar fertilization in grapevines (*Vitis vinifera*) [73]. The shoot application of ALA stimulates the growth of maize seedlings [74]. ALA-based fertilizer can significantly increase the growth of tomato and date palm *(Phoenix dactylifera*) [75,76,77]. Rhizospheric application of ALA significantly increases the endogenous ALA content and improves fruit coloration and interior qualities in apple cultivar (*Malus domestica*) [78].

A dose-effect was observed in the application of ALA in plants, implying that high concentrations of ALA are harmful to plants. For example, low concentrations of ALA (0.5 or 1 mg/L) increase the biomass and the content of various bioactive compounds in oilseed rape, whereas high concentrations of ALA (5 or 10 mg/L) cause oxidative stress, which is detrimental to the growth of oilseed rape [79]. Using this property, ALA is also used as a “photodynamic herbicide” in agriculture [15]. Plants treated with high concentrations of ALA accumulate excess PpIX, which produces ROS in the light, thus causing damage to the plants [80]. Foliar spray of 5 mM ALA resulted in the formation of white necrotic spots on the leaves of rice plants [81]. More interestingly, some herbicides such as diphenyl ether herbicides kill plants by stimulating excessive ALA production, which leads to the accumulation of PpIX in plant cells [82]. As expected, these herbicides are more toxic when used in combination with ALA and enhance weed control [83].

In medicine, ALA is widely used in photodynamic therapy (PDT) [84,85,86], a treatment for certain cancers and precancerous diseases using photosensitizers and lasers. ALA itself is not a photosensitizer, but a precursor substance for the photosensitizer PpIX [87]. When excessive ALA is applied, it is selectively absorbed by proliferating active cells and converted intracellularly into porphyrins such as PpIX. Intracellular PpIX can be activated, leading to ROS generation after exposure to lasers, which is ultimately toxic to tumor cells [88]. Topical ALA-PDT has also been used to treat the tumor or non-tumor dermatoses, which is non-invasive and has better therapeutic effects compared with conventional treatments [89,90,91].

## 5. Conclusions and Future Perspectives

Dramatic global climate change has triggered high temperatures and droughts in some regions and floods in others, seriously threatening crop growth and food security worldwide. Priming significantly improves plant tolerance to various biotic and abiotic stresses, which provides a strategy for improving crop yield and quality in a non-optimal environment [1]. ALA, as a plant growth regulator and a precursor substance of chlorophyll, plays an important role in inducing defense priming [15,16]. Here, we review the remarkable role of ALA priming in plant tolerance. Priming with ALA can activate NO, H_2_O_2_, and hormone signals and strengthen photosynthesis, antioxidant capacity, osmoregulation, and nitrogen assimilation, thus helping plants to obtain a stronger ability to cope with abiotic stresses. High concentrations of ALA function as an herbicide because it causes oxidative stress. Curiously, different plant species have different adaptations to high concentrations of ALA, which may be owing to differences in photosynthetic patterns.

Currently, the use of ALA to activate plant resistance is mainly through seed dips, foliar sprays, and soil watering. The use of operable promoters to construct transgenic plants to produce ALA will facilitate the study of the molecular mechanisms by which ALA regulates plant development and increases plant stress tolerance. Regretfully, most of the current studies on ALA priming have focused on the measurement of physiological parameters; we still know very little about its molecular or biochemical mechanisms. One of the reasons may be that we do not yet dissect the signaling pathways of ALA. For example, is there an ALA receptor? and what are the key components of the ALA signaling pathway? We could address these issues in two ways. (i) Identification of the key components of ALA signaling by screening for mutants with an altered response to ALA through a forward genetic approach. (ii) Identification of ALA-binding proteins (ABP) through biochemical pathways, and then investigation of their functions through a reverse genetic strategy using genome editing techniques. The availability of these genetic materials will help to reveal the physiological functions of ALA and the molecular basis of ALA priming-enhanced plant resistance against numerous environmental stresses.

## Figures and Tables

**Figure 1 ijms-23-00702-f001:**
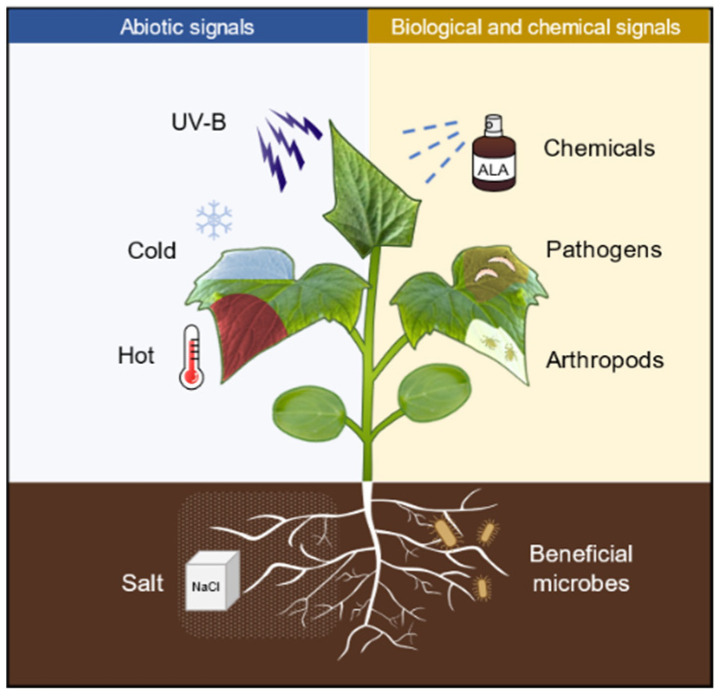
Numerous biotic and abiotic stress as well as defense-related chemicals are capable of inducing plants into priming status. UV-B, ultraviolet B (UVB); ALA, 5-aminolevulinic acid; NaCl, sodium chloride.

**Figure 2 ijms-23-00702-f002:**
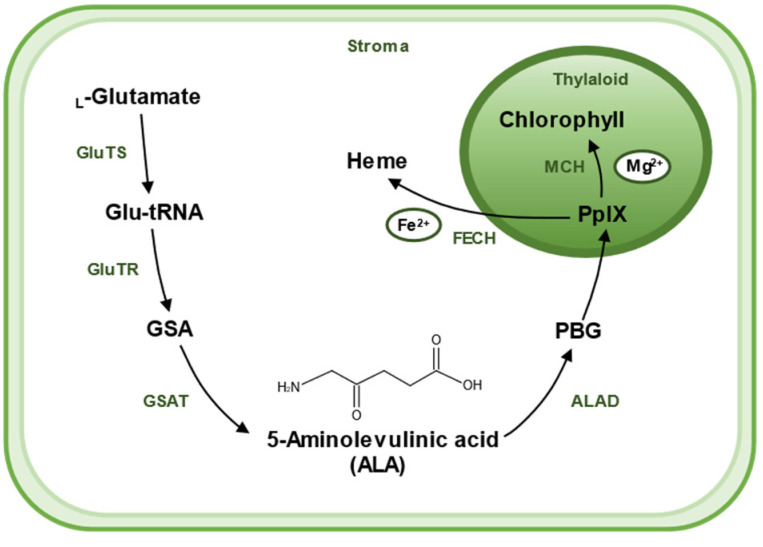
A sketch shows the biosynthetic pathway of ALA and the use of ALA as a substrate for the synthesis of chlorophyll and heme in plants. ALA is created in stroma of chloroplast. The main biosynthetic pathway of ALA is the Beal pathway, which starts from glutamic acid. _L_-Glutamate is ligated to tRNA^Glu^, which is catalyzed by glutamyl–tRNA synthetase (GluTS) to form _L_-glutamy–tRNA. Then, Glu-tRNA is converted to _L_-glutamic acid 1-semialdehyde (GSA), a reaction catalyzed by the key rate-limiting enzyme glutamyl–tRNA reductase (GluTR). GSA then undergoes an isomerization reaction catalyzed by glutamate-1-semialdehyde aminotransferase (GSAT) to form ALA. Two molecules of ALA are catalyzed by ALA dehydratase (ALAD) and agglomerate to form a pyrrole ring called porphobilinogen (PBG). Then, after a six-step enzymatic reaction, four molecules of PBG polymerize to form a porphyrin structure, eventually forming (PpIX). PpIX combines with different enzymes and substrates to yield different products; PpIX chelates Fe^2+^ with Ferrochelatase (FECH) to produce heme, and Mg^2+^ with Mg-chelatase (MCH), and then undergoes a series of catalytic reactions to produce chlorophyll.

**Figure 3 ijms-23-00702-f003:**
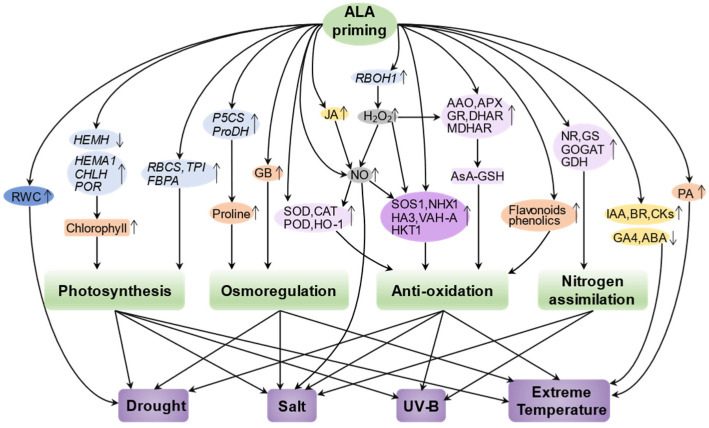
Construction of a regulatory network of ALA priming-mediated abiotic stress tolerance. Priming with ALA enhances the ability of plants to cope with various stresses such as drought stress, salt stress, UV-B stress, and extreme temperature stress by regulating photosynthesis, osmoregulation, antioxidant capacity, and nitrogen assimilation in plants through finely tuning the activities of enzymes (light violet), channel proteins (dark violet), hormones (yellow), signaling molecules (gray), small organic molecules (orange), gene expression (light blue), or physiological levels (dark blue). The upward and downward arrows represent an upregulation or downregulation, respectively. HEMA1, glutamyl-tRNA reductase; CHLH, Mg-chelatase; POR, protochlorophyllide oxidoreductase; NR, nitrate reductase; NiR, nitrite reductase; GS, glutamine synthetase; GOGAT, glutamate synthase; CAT, catalase; SOD, superoxide dismutase; APX, ascorbate peroxidase; GR, glutathione reductase; RWC, relative water content; GDH, glutamate dehydrogenase; P5CS, delta-1-pyrroline-5-carboxylate synthase; HKT1, K^+^ transporter protein 1; NHX1, Na^+^/H^+^ antiporter; VHA-A, proton pump; GB, glycine betaine; DHAR, dehydroascorbic acid reductase; MDHAR, monodehydroascorbic acid reductase; AsA-GSH, ascorbate-glutathione cycle.

**Table 1 ijms-23-00702-t001:** Effects of ALA priming on the tolerance to environmental stressors.

Type of Stress	Plant Species	Stress Concentration	Mode of ALA Application and ALA Level	Effects	References
Salt stress	Asparagus (*Asparagus aethiopicus* L.)	2000 and 4000 ppm NaCl	Foliar application of 3, 5, and 10 ppm	An increase in plant biomass, leaf antioxidant activity, phenolic content, proline accumulation, and photosynthetic rate	Al-Ghamdi et al., 2018
	Barley (*Hordeum vulgare* L.)	150 mM NaCl	Hydroponics of 10, 30, and 60 mg/L	Proline content increased and ROS content decreased	Averina et al., 2010
		100 mM NaCl	Foliar application of 7 ppm	Increased chlorophyll content, antioxidant enzyme activity, and stress responsive gene expression	El-Esawi et al., 2018
	Cassia seed (*Cassia obtusifolia* L.)	100 mM NaCl	Seed soaking of 5, **10**, 15, and 20 mg/L; root irrigation of 10, **25**, 50, and 100 mg/L	Significantly increased chlorophyll content, total soluble sugars, free proline, and soluble protein content; increased photosynthesis and antioxidant enzyme activities	Zhang et al., 2013
	Cucumber (*Cucumis sativus* L.)	75 mM NaCl	Foliar application of 50 mg/L	ALA might delay and counteract the upregulated expression of cucumber PIP aquaporin gene (CsPIP1:1) and cucumber NIP aquaporin gene (CsNIP) genes in cucumber seedlings under NACL stress	Yan et al., 2014
		50 mM NaCl	Foliar application of 25 mg/L	Enhancement of ascorbate-glutathione cycle; increase in shoot and root growth	Wu et al., 2019
		50 mM NaCl	Foliar application of 25 mg/L	Increased ROS production in roots, resulting in upregulation of ion trans-porters SOS1, NHX1, and HKT1	Baral, 2019
		50 mM NaCl	Foliar application of 25 mg/L	Improved plant growth; upregulation of Na^+^/H^+^ antiporter SOS1 and NHX1 at the plasma and vesicle membranes, thereby reducing ion toxicity	Wu et al., 2021
		50 mM NaCl	Foliar application of 25 mg/L	Downregulation of ferrochelatase (HEMH) gene expression; increased in chlorophyll biosynthesis pathway	Wu et al., 2018
	Date Palm (*Phoenix dactylifera* L.)	Seawater treatments at 1, 15, and 30 mS cm^–1^	Root irrigation of 0.08% ALA based fertilizer (PentaKeep-V)	Enhanced photosynthetic assimilation by increasing chlorophyll content and stomatal conductance	Tarek et al., 2007
	Maize (*Zea mays* L.)	100 mM NaCl	Foliar application and seed soaking of 20 mg/L	Improved plant growth; activated the synthesis and accumulation of endogenous NO, thereby increasing the antioxidant capacity of plants	Kaya et al., 2020
	Oilseed rape (*Brassica napus* L.)	100 and 200 mM NaCl	Foliar application of 30 mg/L	Increased plant growth and chloroplast photosynthetic efficiency; reduced Na^+^ uptake and oxidative stress	Naeem et al., 2012
		200 mM NaCl	Foliar application of 30 mg/L	Increased aboveground biomass and net photosynthetic rate; promoted chlorophyll accumulation by promoting increased intermediate levels of the tetrapyrrole synthesis pathway; upregulated the expression of genes P5CS and ProDH encoding proline metabolic enzymes	Xiong et al., 2018
		100 and 200 mM NaCl	Foliar application of 30 mg/L	Improved root and shoot growth; Enhanced plant photosynthesis, chlorophyll content; regulated the uptake of Na^+^ and leaf water potential	Naeem et al., 2010
	Peach (*Prunnus persica* L.)	100 mM NaCl	Foliar application of 200 mg/L	Exogenous ALA treatment could improve the growth and relieve the NACL stress injury of peach seedlings by increasing photochemical efficiency, osmotic content, and antioxidant enzyme activity	Ye et al., 2016
	Radix Isatidis (*Isatis indigotica* Fort.)	100 mM NaCl	Foliar application of 12.5, **16.7**, **25.0,** and 50.0 mg/L	Increased antioxidant enzyme activity, chlorophyll content, and net photo-synthetic rate	Tang et al., 2017
	Sunflower (*Helianthus annuus* L.)	150 mM NaCl	Foliar application of 20, 50, and 80 mg/L	Decreased leaf H_2_O_2_ content and increased SOD activity	Akram et al., 2012
	Swiss chard (*Beta vulgaris* L.)	40 and 80 mM saline (molar ratio NaCl/Na_2_SO_4_ = 9:1)	Foliar application of 60 and 120 μM	The ionic toxicity was reduced by decreasing the Na^+^ content and Na^+^/K^+^ ratio; increased the total nitrogen and GB content	Liu et al., 2014
	Watermelon (*Citrullus lanatus* L.)	100 mM NaCl	Foliar application of 1.25 mM	Regulated nitrogen metabolism, reduced ion toxicity caused by salt stress, and increased soluble protein and proline	Chen et al., 2017
Extreme Temperature	Cucumber (*Cucumis sativus* L.)	42/38 °C (day/night)	Foliar application of 3 μM	Reduced ROS content and growth inhibition under heat stress; enhanced antioxidant enzyme (SOD, CAT, and GPX) activity and proline content	Zhang et al., 2012
		12 °C/8 °C (day/night)	Foliar application of 15, **30**, and 45 mg/L ALA	Nutrient contents (N, P, K, Mg, Ca, Cu, Fe, Mn, and Zn) and endogenous hormones (JA, IAA, BR, IPA, and ZR) were enhanced in roots and leaves; Increased chlorophyll content, photosynthetic capacity, and antioxidant enzymes (SOD, POD, CAT, APX, and GR); reduced growth inhibition of seedlings by cold stress	Anwar et al., 2018
		16 °C/8 °C (day/night)	Add to the culture substrate of 10, 20, or 30 mg ALA·kg^−1^ (ALA were mixed with a constant weight of substrate (kg))	Significantly reduced plant growth inhibition; increased chlorophyll content, antioxidant enzymes (SOD, CAT, and POD) activity; reduced accumulation of ROS and malondialdehyde in roots and leaves	Anwar et al., 2020
	Drooping wild ryegrass (*Elymus nutans* Griseb.)	5 °C	Seed soaking of 0.1, 0.5, **1**, 5, 10, and 25 mg/L	Significantly increased seed respiration rate and ATP synthesis and protected germinating seeds from cold stress; increased GSH, AsA, total glutathione, and total ascorbate concentrations, as well as SOD, CAT, APX, and GR activities	Fu et al., 2014
		5 °C	Foliar application of 0.5, **1**, 5, 10, and 26 mg/L	NO might be a downstream signal that mediates ALA-induced cold tolerance, thereby enhancing antioxidant defense	Fu et al., 2015
		5 °C	Root soaking of 0.5, **1**, 5, 10, and 26 mg/L	NO might act as a downstream signal to mediate ALA-induced cold resistance by activating antioxidant defense and PM H^+^-ATPase and maintaining Na^+^ and K^+^ homeostasis	Fu et al., 2016
	Maize (*Zea mays* L.)	14 °C/5 °C (day/night)	Foliar application of 0.15 mM	Increased proline accumulation, antioxidant enzymes (SOD and CAT) and RuBP carboxylase activity; prevented reductions in maize crop yield due to low-temperature stress	Wang et al., 2018
	Red pepper (*Capsicum annuum* cv. Sena)	4 °C	Seed soaking of 1, 10, **25**, **50**, and 100 ppm	Resulted in higher germination and seedling emergence percentages, as well as faster germination and seedling emergence	Korkmaz et al., 2009
		3 °C	Seed soaking, foliar spray and soil drench of 1, 10, **25**, **50**, and 100 ppm	Improved plant quality, chlorophyll content, sucrose, and proline content; enhanced SOD activity	Korkmaz et al., 2010
	Rice (*Oryza sativa* L.)	3 °C, 5 °C	Root soaking of 0.001, 0.1, 1, and 5 ppm	Reduced cold injury-induced tissue electrolyte leakage	Hotta et al., 1998
		10 °C	Seed soaking of 8.5 mM	Increased antioxidant enzymes (SOD, POD, APX, and GPX) activity; increased relative gene expression of enzymes of PA biosynthesis	Sheteiwy et al., 2017
	Soybean (*Glycine max* L.)	4 °C	Hydroponics of 5, 10, 15, 20, 30, and 40 μM	Increased chlorophyll content and relative water content of leaves; enhanced activity of antioxidant enzymes CAT and HO-1	Balestrasse et al., 2010
	Tomato (*Solanum lycopersicum*)	15 °C/8 °C (day/night)	Foliar application of 5, 10, **25**, 50, and 100 mg/L	ALA induced H_2_O_2_, which in turn increased the ratio of GSH and ASA, leading to enhanced antioxidant capacity; significantly increased the activities of SOD, CAT, APX, DHAR, and GSH	Liu et al., 2018
		15 °C/8 °C (day/night)	Foliar application of 5, 10, **25**, 50, and 100 mg/L	ALA pretreatment-induced CAT and ASA-GSH reliably eliminated excess ROS under low temperature stress and maintained redox homeostasis	Liu et al., 2018
		15 °C/8 °C (day/night)	Foliar application of 25 mg/L	ALA triggered NO production directly, or induced H_2_O_2_ and JA signals to trigger NO production, thus NO interacted with JA to regulate cold-induced oxidative stress	Liu et al., 2019
Drought stress	Kentucky bluegrass (*Poa pratensis* L.)	10% PEG 6000	Foliar application of 10 mg/L	Improved turf quality and leaf relative water content; enhanced antioxidant enzymes (SOD, CAT, APX, GPX, DHAR, and GR), ASA, and GSH content, thus reducing oxidative damage	Niu et al., 2017
	Oilseed rape (*Brassica napus* L.)	Drought stress (40% of water-holding capacity)	Foliar application of 30 mg/L	Maintained relatively higher leaf water status; enhanced chlorophyll content and net photosynthetic rate; increased antioxidant enzyme (POD and CAT) activity	Liu et al., 2013
		Drought stress (40% of water-holding capacity)	Foliar application of 30 mg/L	Expression of photosynthetic genes (RBCS, TPI, FBP, FBPA, and TKL) was upregulated; increased leaf hexose and sucrose accumulation and maintenance of starch content	Liu et al., 2016
	Sunflower (*Helianthus annuus* L.)	Water stress (70% field capacity)	Foliar application of 10, 20, and **30** mg/L	Reduced oxidative damage by lowering H_2_O_2_ and MDA contents	Rasheed et al., 2020
		Drought stress (40% of water-holding capacity)	Foliar application of 25, 50, **75**, and 100 mg/L	Enhancement of stay green and CAT, SOD, and APX activities, thus reducing drought-induced yield losses and improving oil contents	Sher et al., 2021
	Wheat (*Triticum aestivum* L.)	Irrigation interval of 7, 14, and 21 days	Foliar application of 25, 50, and **100** ppm	Increased grain yield	Al-Thabet et al., 2006
		Water deficit (60% and 80% of field capacity)	Foliar application of 50, **100**, and **150** mg/L	Improved leaf fluorescence (qN, NPQ, and Fv/Fm), shoot and root K^+^, root Ca^2+^, proline, and GB accumulation	Akram et al., 2018
		Water stress (30% maximum water capacity)	Foliar application of 30 mg/L	Increased plant growth, photosynthesis, and chlorophyll content; reduced the degree of damage to cell membranes during early nutritional development	Ostrowska et al., 2019
		80% (mild drought stress), and 60% (high drought stress)	Foliar application of **50**, **100**, and 150 mg/L	Increased fresh and dry weight of shoots and roots, chlorophyll content, GB content, and N content in leaves and roots	Kosar et al., 2015
UV-B stress	lettuce (*Lactuca sativa* L.)	3.3 W m^−2^ UV-B	Foliar application of 10 and 25 ppm	ALA treatment resulted in a substantial increase in phenylalanine ammonia lyase (PAL) and γ-tocopherol methyltransferase (γ-TMT) gene expression, antioxidant enzyme activity, and chlorophyll a and b concentrations.	Aksakal et al., 2017
	Pigeon pea (*Cajanus cajan* L.)	enhanced UV-B (2.2 kJ m^−2^d^−1^)	Seed soaking of 25 and **100** μM	Reduced germination time and increased germination index; upregulated photosynthesis, antioxidant enzymes (CAT, SOD, and POD), total phenolic content, and total flavonoid content to balance ROS and reduce UV-B damage to plant productivity	Gupta et al., 2021
		enhanced UV-B (2.2 kJ m^−2^d^−1^)	Seed soaking of 25 and 100 μM	Increased plant growth and growth regulating parameters; increased enzyme activity and non-enzymatic antioxidant content in the plant defense system and reduced oxidative stress in seedlings	Gupta et al., 2021

The optimum levels of ALA are shown in bold.

## Data Availability

Not applicable.

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
