# Peer review of "Advances in 5-Aminolevulinic Acid Priming to Enhance Plant Tolerance to Abiotic Stress"

_ijms, 2022, doi:10.3390/ijms23020702_

Round 1

Reviewer 1 Report

This manuscript is written to demonstrate the advances in 5-aminolevulinic acid priming to enahnce plant tolerance to abiotic stress. The topic is good and up-to-date, however this review type MS cannot fulfill this goal to give a really nice overview of function and priming response of interesting molecule, ALA. Many places authors just write some general sentences not mining deeply the exact mechanism of these responses, eg. authors mentioned generally "in plants" and not selectively which plants, in which developmental stages, resistant or sensitive, etc. Big problem is the lack of those experiments which studied the ALA effects in mutant or transgenic plants with modified ALA, the lack of summarizing tables of plants treated with exogenous ALA, etc.

In this form, this manuscript is recommended to be rejected.

Author Response

Response: Many thanks for your constructive suggestions and comments. As requested, we had comprehensively revised the manuscript by adding more detailed information on the plant species, developmental stages and the priming effects of ALA. To better present this information, we had added a table.

Reviewer 2 Report

The authors present here a review paper on Advances in 5-Aminolevulinic Acid Priming to Enhance Plant Tolerance to Abiotic Stress. This is indeed a topic of high interest to many readers. However, currently, there are a fairly large number of review articles one way or another relating to this topic (e.g. 5-Aminolevulinic acid (ALA) biosynthetic and metabolic pathways and its role in higher plants: a review; 5-aminolevulinic acid-mediated plant adaptive responses to abiotic stress). Unfortunately, the research community is still far from understanding the real mechanisms involved in the process. This review contains interesting and useful facts, however, the authors failed to provide more in depth details of the studied mechanisms how 5-Aminolevulinic acid priming improves plant tolerance to environmental stresses. The authors need to provide more critical analysis on the mechanisms involved. Beside this, the paper is lacking a good graphical presentation. The authors need to provide 2-3 more related figures and a table with different species of plants treated with 5-Aminolevulinic Acid for stress tolerance.

Author Response

Response: Many thanks for your constructive suggestions and comments. As requested, we had added a table providing comprehensive information including types of environmental stressors, plant species, experimental results and references. We had comprehensively and thoroughly revised this manuscript to provide new insights into the priming mechanisms of ALA on enhancing tolerance against various stresses.

Round 2

Reviewer 1 Report

I accept all the revisions and I think the quality of this manuscript is good for publication in IJMS.

Author Response

Response: We highly appreciate your suggestions and comments, which have greatly improved the quality and readability of this manuscript.

Reviewer 2 Report

The revised version of the MS is much improved however, the MS still lacking good graphical presentation. The authors need to provide a figure for section 3.  

Also check and revise this sentence ''. Interestingly, besides improving the
drought resistance of plants, priming with ALA can also improve the waterlogging tolerance, with higher levels of antioxidant enzyme activity, photosynthetic efficiency, and root respiration in Fig (Ficus carica) [68].

Author Response

Response: As requested, we separated the original images and put them in the second and third section respectively, while we added a new image in the first section. These images could better present the main content of this overview.

As suggested, “Interestingly, besides improving the drought resistance of plants, priming with ALA can also improve the waterlogging tolerance, with higher levels of antioxidant enzyme activity, photosynthetic efficiency, and root respiration in Fig (Ficus carica)[68].” had been revised to “Interestingly, in addition to enhance the drought resistance of plants, ALA priming also improves waterlogging tolerance in Fig (Ficus carica), with higher levels of antioxidant enzyme activity, photosynthetic efficiency, and root respiration[68].”